# The History and Prediction of Prebiotics and Postbiotics: A Patent Analysis

**DOI:** 10.3390/nu16030380

**Published:** 2024-01-27

**Authors:** Tao Zang, Lu Han, Zhaoxiang Lu, Lulu Tan, Dunsheng Liang, Xiaofan Shen, Xiaoping Liao, Yahong Liu, Hao Ren, Jian Sun

**Affiliations:** 1Guangdong Laboratory for Lingnan Modern Agriculture, State Key Laboratory for Animal Disease Control and Prevention, College of Veterinary Medicine, South China Agricultural University, Guangzhou 510642, China; zangt1122@163.com (T.Z.); luhan@scau.edu.cn (L.H.); lucaszhaoxiang@gmail.com (Z.L.); tllnnd@stu.scau.edu.cn (L.T.); liangdunsheng@stu.scau.edu.cn (D.L.); whpusxf@163.com (X.S.); xpliao@scau.edu.cn (X.L.); lyh@scau.edu.cn (Y.L.); hao.ren@scau.edu.cn (H.R.); 2Guangdong Provincial Key Laboratory of Veterinary Pharmaceutics Development and Safety Evaluation, South China Agricultural University, Guangzhou 510642, China; 3Jiangsu Co-Innovation Center for the Prevention and Control of Important Animal Infectious Diseases and Zoonoses, Yangzhou University, Yangzhou 225000, China

**Keywords:** prebiotics, postbiotics, patent analysis, gut microbiota

## Abstract

Prebiotics and postbiotics have gained attention as functional food additives due to their substantial influence on the gut microbiome and potential implications for human health on a broader scale. In addition, the number of patents for these additives has also increased, yet their functional classification has been problematic. In this study, we classified 2215 patents granted from 2001 to 2020 by functionality to enable predictions of future development directions. These patents encompassed subjects as diverse as feed supplementation, regulation of intestinal homeostasis, prevention of gastrointestinal ailments, targeted drug administration and augmentation of drug potency. The progression of patents issued during this time frame could be divided into three phases: occasional accounts prior to 2001, a period from 2001 to 2013 during which an average of 42 patents were issued annually, followed by a surge exceeding 140 patents annually after 2013. The latter increase has indicated that pre- and post-biotics have been recognized as biologically relevant. Patent mining therefore can enable forecasts of the future trajectory of these biologics and provide insights to evaluate their advancement. Moreover, this research is the first attempt to generalize and predict the directions of prebiotics and postbiotics using patent information and offers a comprehensive perspective for the potential utilization of prebiotics and postbiotics across a wide variety of fields.

## 1. Introduction

Prebiotics derived from non-digestible carbohydrates exhibit a selective capacity to stimulate the metabolism and proliferation of beneficial bacteria thereby improving the composition of the gut microbiota [1,2]. In 1995, Gibson and Roberfroid identified prebiotics as a class of plant-derived oligosaccharides that exhibit selective activity towards beneficial targets in the intestinal flora such as *Bifidobacterium* and *Lactobacillus.* Prebiotics provide nutrients and other functions to gut microbiota [2] and encompass a wide range of substances including oligosaccharides composed of fructose (FOS) such as inulin, galactose (GOS), isomaltose (IMO), xylose (XOS) as well as lactulose, polydextrose, lactitol, vegetable protein hydrolysates, Chinese herbal medicines and wild plants [3,4]. Prebiotics generally exhibit thermal and acid tolerance and decreases the relative abundance of pathogenic bacteria and choline metabolizers [5].

Postbiotics are derived from metabolites or cell wall fragments generated via probiotics. It is a new term that refers to a wide range of bioactive molecules including non-viable or inactivated microbial cells, short-chain fatty acids, vitamins, enzymes, teichoic acid, peptides and exopolysaccharides derived from beneficial microorganisms [6,7]. Metabolic byproducts are currently employed as functional ingredients in dairy products that directly or indirectly mediate positive biological activities. For instance, peptides and exopolysaccharides exert multiple effects including immune enhancement, intestinal flora homeostasis and regulation of physiological functions [8]. The survival and stability of probiotics in dairy products is subject to various processing conditions such as low pH, temperature as well as the matrix composition including the levels of fat, protein, carbohydrates, natural antimicrobials and water activity. In contrast, prebiotics are stable and present numerous technological advantages over probiotics in dairy products and their integration into dairy products can yield novel functional products [9,10,11]. This is especially useful considering their wide range of storage conditions and possess a vast potential for future applications.

The mammalian gastrointestinal tract harbors a substantial microbial ecosystem with an estimated 10^11^ to 10^12^ microorganisms [12]. This microflora is essential for intestinal endocrine function, immune cell maturation, protection against pathogen overgrowth, cancer inhibition, reduction in cardiovascular disease and prevention of obesity [13]. Prebiotics and postbiotics released from non-digestible compounds via fermentation facilitate the communication between the microbiota and the host [14] and especially have a positive effect on the regulation of colonic epithelial cell permeability [15]. Moreover, polysaccharide-based electro-spun fibers have been used to encapsulate prebiotics to enhance their colon-specific targeting properties, thereby augmenting their human colon cancer inhibitory properties [16]. Similarly, obesity control using both prebiotics and postbiotics have shown success for long-term weight control [17]. Importantly, pre- and post-biotics have a significant effect on the gut microbiota and human health [18].

Pre- and post-biotics have also shown homeostatic effects that balance intestinal microorganism populations to improve human health [19]. Recently, their combined effects on the host intestine have resulted in increasing numbers of patents although the functional categorization of these related patents is confusing [1]. In this study, we examined 2215 prebiotics and postbiotics patents that were granted from 2001 to 2020 as a reflection of scientific and technological development. And we used these to identify the most active research areas and predict future development trends for these biologics [20]. This research represents the first attempt to predict the development direction of prebiotics and postbiotics through patent mining, thereby holding a significant reference value in evaluating development of these compounds. This analysis provides a comprehensive framework for understanding of the development trends and future application fields for prebiotics and postbiotics.

## 2. Materials and Methods

### 2.1. Data Collection

This study conducted a patent search related to prebiotics and postbiotics from 2001 to 2020, using the following databases for patent retrieval and analysis: Innography (https://app.innography.com/), accessed on 8 November 2022. Yi patent search and analysis (https://patents.justia.com/inventor/yi-yang/, accessed on 8 November 2022), Agricultural and Natural Resources from CABI (https://www.cabdirect.org/, accessed on 8 November 2022), Web of Science (https://clarivate.com, accessed on 8 November 2022), China Knowledge Resources Database (https://www.cnki.net/, accessed on 8 November 2022), International Agricultural Science and Technology Information System from the Food and Agriculture Organization of the United Nations (https://www.fao.org/agris/, accessed on 8 November 2022) and the Agricultural Online Search Database from the National Agricultural Library (https://www.nal.usda.gov/, accessed on 8 November 2022). In particular, the Innography database was a powerful tool for patent value judgment and analysis and it contains more than 90 million patents from over 100 countries and regions around the world including US patent litigation, trademark and organizational business data. Noticeably, its unique patent strength index can be used to mine core patents and the text clustering function can quickly analyze distributions of patented technologies.

### 2.2. Search Queries and Classification Codes

The search results were making more comprehensive and accurate through the use of International Patent Classification (IPC) numbers and word exclusion (Table 1). The Boolean Operator was designed as TS = (((postbiotics or probiotics or prebiotics or lactobacillus or “microbial ecological” or “microbial pharmaceutics” or “microecology preparation” or “micro-organism regulator” or “microbial agents” or “microbiological agents” or “beneficial bacteria” or “lactic acid bacteria” or bifidobacterium or bifidobacteria or “bacterium bifidum” or “streptococcus lactis” or lactobacillus or lactobacilli or “lactococcus lactis” or “Streptococcus lactis” or “clostridium butylicum” or bacillus or saccharomyces or yeast or “Saccharomyces cerevisiae” or Cerevisiae or Ferment or Acidophilus or Rhamnobacteria or “Carboxybutyric acid bacteria” or “Clostridium butylicum” or “C. butyricum” or Clostridium or Micro-ecological or environment or microbe or bacterial or Microorganisms or Micro-organisms or Microbe or Microbiology or Colon-specific or “Colonic specificity” or “Specific site intestine” or “Specific site colon” or “Specific site colorectal” or Flora or “Intestine functionality” or “Directional release intestinal”) not (non-intestinal or skin or injection or “Tyrosine hydroxylase” or “tyrosine kinase”)) and @* (ipc_A23K001 or ipc_A23K003 or ipc_A23K010 or ipc_A23K020 or ipc_A23L031 or ipc_A23L033 or ipc_A61K031 or ipc_A61K035 or ipc_A61K009 or ipc_A61K038 or ipc_A61P001 or ipc_A61P003)), and the symbol “@*” was set to present the cognate words of prebiotics and postbiotics and aims to output any of the patents containing the words and function of “prebiotics and postbiotics”. Table 1 shows an overview of combinations with keywords and research contents derived into the final patent dataset. These searches query was expanded, and these relevant patents were assessed based on the title and abstract.

Research manuscripts reporting large datasets that are deposited in a publicly available database should specify where the data have been deposited and provide the relevant accession numbers. If the accession numbers have not yet been obtained at the time of submission, please state that they will be provided during review. They must be provided prior to publication.

Interventionary studies involving animals or humans, and other studies that require ethical approval, must list the authority that provided approval and the corresponding ethical approval code.

### 2.3. Data Analysis

Statistical analysis and graphing were conducted using Prism software V. 8 (GraphPad, Boston, MA, USA). Network diagrams were created with Cytoscape V. 3.9.0 (https://cytoscape.org/, accessed on 8 November 2022) drawing upon the patent analysis results using Boolean Operators. In addition, priority countries and institutions were collected and counted on patent grant years while other countries in the same patent family were used as statistics for technology transfer analysis. Statistical analyses were conducted on technology transfer in the same patent family from other countries.

## 3. Results and Discussion

### 3.1. Global Distribution of Prebiotic and Postbiotic Patents

#### 3.1.1. Total Number of Prebiotics and Postbiotics Worldwide

Prebiotics and postbiotics are frequently utilized to regulate gut health and augment immune responses, and due to their efficacy, they are currently receiving increased attention. To understand the development trend of prebiotics and postbiotics, a total of 2215 global prebiotics and postbiotics patents were searched through the above database from 2001 to 2020. The patents of prebiotics and postbiotics were primarily classified as feed additives, prevention and treatment of gastrointestinal diseases, targeted delivery of drugs and enhancement of drug effects. These classifications indicating that the pre- and post-biotic functionalization were innovative. For instance, prebiotics were used as aids to provide protection against COVID-19 active infection by minimizing inflammation [15]. This demonstrated the extraordinary application potential of prebiotics. The numbers of prebiotics and postbiotics patents by country were as follows: United States (447), China (286), European Patent Office (EPO, 230), Japan (200), Canada (136), Australia (120), South Korea (82), Brazil (71) and other countries and organizations (1088) over the last two decades (Figure 1a). Recent studies have reported on roles for prebiotics in modulation of the gut-brain and gut-liver axes, immunomodulation and cancer [21,22]. Consequently, an increasing number of patents for prebiotics and postbiotics are emerging necessitating patent analysis. The number of patents increased by 1.45 percent from 2011 to 2020 compared to 2001 to 2010.

#### 3.1.2. Annual Patents Applications Worldwide

In view of total annual patents worldwide and their distribution in different countries, the annual patents in different countries were analyzed from 2001 to 2020 (Figure 1b). The global patent technology could be roughly divided into three stages. The first phase was from 2000 to 2006 and globally, annual patent applications grew from 42 to 159. These applications were primarily from the United States, EPO, the World Intellectual Property Organization (WIPO) and Canada. The second phase was from 2007 to 2012, and the average annual number of patent applications remained at about 100 and were primarily from the United States, EPO, WIPO, China and Japan. The third stage was after 2013 and the number of patent applications increased again and the average annual patent application number was >140 and originated in the United States and China, Japan, the WIPO and the EPO. After 2013, research on patented technologies for prebiotics and postbiotics emerged, and global research centers have spread globally, with China and Japan displayed increased development. In this period, prebiotics and postbiotics have been increasingly used in treatment of metabolic, inflammatory and immune disease, colon cancer, and a corresponding number of patents have been approved [23]. Undoubtedly, prebiotics play a crucial role in immune regulation, anti-tumor activity, promotion of mineral element absorption and regulation of fat metabolism. Prebiotics have significantly increased intestinal populations of beneficial microorganisms [24]. Previous studies have demonstrated that prebiotic metabolic regulation via beneficial microorganisms in the gastrointestinal system included lactobacilli and bifidobacteria [25]. Furthermore, prebiotics and postbiotics act on the gut microbiome and have been regarded as ‘a second brain’. These patents are expected to spur on the rapid development of non-antibiotic therapies for the gut microbiome.

#### 3.1.3. Trends in Worldwide Patent Applications

To understand the evolution of patent applications over time, the annual trends in patent applications were investigated. Notably, the life cycle curve composition of annual patents applications showed a reciprocal structure indicating many technological innovations have taken place in this field. The most obvious of these occurred twice (2006 and 2013) and caused a surge in patent numbers in this field. Despite a decline in patent applications from 2017 to 2018, the overall level remained relatively high, potentially indicating an impending wave of innovation (Figure 1c).

Gradual year-by-year increases in the application trends of prebiotic and postbiotic patents were evident from 2001 to 2020. The extensive utilization of prebiotics and postbiotics across various domains including medicine, food, health care, animal husbandry and related sectors has resulted in substantial prospects for growth potentially playing a role in the escalation of pertinent patent filings. The prebiotic market is projected to grow at a rate of 12.7 percent over the next eight years [26].

### 3.2. Geographical Analysis of Patent Applications

#### 3.2.1. Analysis of Global Patent Geographical Distribution

The global distribution of prebiotics and postbiotics patents could be assessed according to the primary countries contributing to technological advancements in the field. The technologically developed countries were led by the United States, Japan and Germany. In particular, 955 patent applications were filed by American inventors, and 230 were filed in the US Patent and Trademark Office while 725 were foreign patents. These foreign patents were filed primarily in the EPO and WIPO but also in Japan, China, Canada, Brazil, Australia and Mexico. Likewise, Australia and Japan have applied for 43 and 41 patents in domestic patent applications while foreign applications accounted for 65.2 and 77.4 percent, respectively. Moreover, foreign applicants attach importance to the layout of patents in this field. However, China applied only 61 patents abroad (29.2 percent) indicating a weaker awareness of patent distribution among Chinese applicants (Figure 2a). Overall, the development of these new technologies and methods presents promising prospects for prebiotics and postbiotics and can enable real-time assessments of human subjects to quantify their health levels, thus propelling the field forward [27]. Therefore, prebiotics and postbiotics patents could facilitate the development of novel technologies.

#### 3.2.2. Analysis of Patent Technology Source Countries

To gain further insight into the inventor locations of patent technology, we analyzed the data by source country since this can also reflect the innovation level of a country or region. Five countries generated the bulk of the applications and included the United States, China, Germany, Ireland and the United Kingdom. The United States applied for 955 patents and represented 43.1 percent of the total global applications revealing as the innovation center of prebiotics and postbiotics. The developed countries represented by the United States and Europe were the primary technological innovators while France, Australia, Switzerland, Japan, Belgium, Canada, South Korea, the Netherlands, Italy, Austria and Israel and others filed < 75 patent applications. Applications from highly developed economies were centered in pharmaceutical industries indicating a close association with pharmaceutical and economic development. Most importantly, China has rapidly emerged as a major player ranking second in the world for prebiotics and postbiotics patent applications highlighted promising development prospects (Figure 2b). Chinese research on prebiotics and postbiotics is currently expanding especially since current research indicates that postbiotics can mediate beneficial effects on the microbiome and these include proteins, peptides, organic acids and other small molecules [28]. A method for prebiotic encapsulation holds a great potential as an effective formulation for colon cancer prevention [16].

#### 3.2.3. Analysis of Patented Technology Application Countries

Technology applications were primarily filed in the United States (448) while China, EPO, WIPO, Japan, Canada and Australia each filed ~100. In addition, these technologies have spread from the country of origin indicating patent layouts in all major technologically developed countries. For example, the number of patent applications accepted by China’s national intellectual property rights were 291, far higher than the 209 patent applications filed by Chinese inventors. Patent layouts create conditions for its products to enter China indicating that China will face strong market competition in the field of prebiotics and postbiotics. Apart from China, Canada, Australia, Brazil, Mexico and India were also significant hubs for patent applications. Despite lacking industrialization comparable to Japan, United States and Europe, these countries have successfully received patents according to the extensive scope of their markets. For instance, the major health care companies Leli, Beton, Brighten, Bayer and By-Health have introduced a wide range of prebiotic beverages, fruit and vegetable enzyme supplements, prebiotic dietary fiber and jellies, postbiotic skin care products and milk powder. The development and application of patented technology have benefitted these products and have become ubiquitous in the retail marketplace.

#### 3.2.4. Analysis of China Foreign Patent Applications

In view of the status of foreign patent applications in China, an analysis was conducted on the dissemination of Chinese inventions and creations through overseas patents. Among the 209 patent applications filed by Chinese inventors, 148 were filed with the State Intellectual Property Office (SIPO) of China and 61 were filed abroad. The distribution of applications in other countries and regions was as follows: WIPO (14), U.S. Patent and Trademark Office (10), EPO (6), Canada (5), Australia (3), Japan Patent Office (2), Mexico (2) and Brazil (1). In contrast, the Patent Cooperation Treaty (PCT) route was the primary means for Chinese applicants to apply for patents. However, the PCT was only a patent application route. To obtain exclusive rights in international markets, China foreign patent applications were necessary to proceed with the national phase and obtain authorization. Therefore, China was granted only 21 patents with valid patent rights including 4 in the United States, 3 in Canada, 2 each in Hong Kong, Australia, Russia, Japan, South Africa, and Europe and 1 each in Mexico and South Korea. Chinese applicants generally exhibited weaker intellectual property risk management in this field (Figure 2d).

#### 3.2.5. Ranking of Patent Provinces and Cities in China

China inventor attribution statistics were used to clarify the patented technologies of prebiotics and postbiotics across different provinces and municipalities in China. This analysis revealed that Guangdong Province exhibited the highest number of patent applications followed by Zhejiang, Jiangsu, Beijing, Shanghai and Shandong Provinces each with >10 patent applications. Additionally, Hunan, Liaoning, Anhui, and Sichuan Provinces also secured positions within the national top 10 list (Figure 2e).

### 3.3. Patent Applicant Analysis

#### 3.3.1. Global Applicant Ranking Analysis

We also produced an overview of the ranking and distribution of prebiotics and postbiotics applications in global companies and individuals. The Groupe Danone SA emerged as the leading applicant for patent filings in this domain followed by Nestle SA and Sigmoid Pharma Limited. In particular, Shanghai Jiaotong University ranks among the top 20 global patent applications with 22 patent applications (Figure 3a). Of note, the prebiotic and postbiotic patents of these companies and individuals were primarily utilized in food, medicine, feed additives and pharmaceutical preparations. Prebiotics and postbiotics have been identified as potentially significant contributors in the prevention of oxidative stress-related diseases induced by reactive oxygen species that are linked to cancer, neurodegenerative and cardiovascular disease as well as diabetes. Furthermore, as this technology continues to progress, utilization of prebiotics and postbiotics is expected to expand into more sophisticated domains offering potential solutions to various human crises.

#### 3.3.2. Global Applicant Competitiveness Analysis

A view of relationships between applications, authorizations and global market competitiveness of major global applicants was also generated for prebiotics and postbiotics (Figure 3b). The larger the bubble in the right half of the graph, the stronger the advantage in pa-tent applications. Group Danone SA exhibited the highest level of patent technology competitiveness and was followed by Nestle SA, Sigmoid Pharma Limited, Valeant Pharmaceuticals International, Cedars Sinai Medical Center and Seres Therapeutics. These entities are positioned in the right half of the bubble chart indicating their robust technical competitiveness. Notably, Nestle SA positioned above the bubble chart demonstrated strong economic strength. Among the top 20 patent applicants in the field of prebiotics and postbiotics, the Massachusetts Institute of Technology (MIT) was the sole university. A relative mature market competition in prebiotics and postbiotics dominated by enterprises and universities and research institutions can collaborate with enterprises to effectively transform scientific and technological achievements. Despite ranking second in worldwide patent applications, the institutional competitiveness of Chinese institutions appears to be weak and Chinese institutions were not listed among the top 20 global competitors. This result may be attributed to fragmented Chinese applicants, lower patent grant rates and a lower number of cognate patents.

### 3.4. Analysis of Prebiotic and Postbiotic Relevant Patents

Patents for metabolic regulators were also queried (see Section 2.1) to determine the status of research of pharmaceutical agents (Figure 4a). Text clustering diagrams were generated where the inner circle represented primary technical themes distinguished by color. The outer circle encompassed technical points associated with the inner circle. Specifically, the primary research subjects were probiotic, fatty acids, lactic acid, amino acid, polyethylene glycol and glycan. Prebiotics were primarily associated with microcapsules, active substance, microparticles and dietary fiber while fatty acids predominantly encompassed short-chain fatty acids. Lactic acid was linked to lactic acid bacteria whereas amino acids were involved with growth factors, fusion proteins and similar entities. Polyethylene glycol was primarily associated with research on pharmaceutical formulations and protective coatings (Figure 4b).

The primary patented technologies of prebiotics and postbiotics encompassed enteric coatings, dosage forms, therapeutic agents, delivery systems, water-insoluble materials and sustained release mechanisms. Specifically, enteric coating involved utilization of methacrylic acid while dosage forms encompassed active pharmaceutical, pharmaceutical dosage form, diagnostic agents, polymer shells and bead populations. Therapeutic agent was linked to gastric, covalently linked compounds and protective side elements. The delivery system primarily encompassed organic acid, colonic delivery, drug and oral delivery systems, cargo molecule and biologically active agent. Water-insoluble was grouped with core material, water-insoluble polymer, clothing film, digestion liquid and intestinal digestion while sustained release primarily involved acceptable salts and selective serotonin (Figure 4c). Prebiotics and postbiotics are able to modulate the action of hepatic lipogenic enzymes by elevating organic acid production [5]. However, prebiotics can be affected by temperature, pH and chemical stressors that can alter their complete transport and activity in the colon [29]. Therefore, the most active areas of research were delivery systems and dosages.

Association network diagrams were constructed by considering the frequency of prebiotics and postbiotics in conjunction with other metabolites (Figure 5). The section of patents related to prebiotics, postbiotics, and other metabolites is in the Appendix A. Specifically, the network diagram of the association between prebiotics and metabolites were primarily related to flora, metabolite, intestinal flora, fatty acid and polyethylene glycol (Figure 5a). The multi-level relationship between prebiotics and postbiotics and other metabolites indicated that they are closely related to proteins, amino acids, lactic acid and polysaccharides. The network diagrams indicated that prebiotics play an important role in the animal body with a variety of regulatory mechanisms (Figure 5b). Prebiotics and postbiotics are closely linked to organic acids, lactic acid, nicotinic acid and metabolism, which was consistent with existing research [30]. In the foreseeable future, forthcoming areas of focus for prebiotic therapy encompass hypercholesterolemia [31] and obesity [24], colorectal cancer [32], mood disorders [33] as well as the modulation of the brain-gut-microbiome axis [34].

## 4. Discussion and Limitations

Here, we show that probiotics and prebiotics are currently stabilising patent applications for research in veterinary gut health. Patents for probiotics and feeds are most prominent in North America and Europe, followed by China. Patent applications for prebiotics and postbiotics have far-reaching implications for future research [35]. Firstly, the number and quality of patent applications can reflect the quantity and frontier research in this field, which provided an important reference information for researchers [36]. Secondly, patent applications can protect intellectual property of researchers and technological innovations and these patents enable them to gain economic benefits from their patents. Patent applications perhaps attract more funds and talents to investigate in this field [37]. Thirdly, patent applications can also promote market competition in order to maintain a sustainable advantage in innovation patents [38]. This will contribute to the advancement of the prebiotics and postbiotics market, thereby creating additional application fields and business opportunities for future research.

Probiotics, prebiotics and postbiotics have been used in a wide range application. In the field of health foods and nutritional supplements, they can improve intestinal microbial communities, enhancing immunity, promoting digestion and absorption, and preventing and treat some diseases [39]. In the field of animal farming, they can maintain the balance of intestinal microflora, increasing the utilisation of feed, enhancing the disease resistance of animals and improving the quality of meat and eggs [40]. Besides, prebiotics and postbiotics can be used to treat intestinal diseases, regulating the immune system and improving cardiovascular health [15]. In addition, the widespread use of prebiotics and postbiotics in food possess several remarkable advantages, especially compared to the direct use of probiotics. Firstly, the stability of prebiotics and postbiotics is superior than probiotics, which are less susceptible to processing and storage conditions such as temperature, humidity and pH [41]. Prebiotics and postbiotics were used to a wide range of food products and maintaining a longer shelf-life. Secondly, prebiotics and postbiotics are various bioactive substances derived from probiotics, and these live bacterial preparations possessing the potential to carry virulence and antibiotic resistance genes [42]. Finally, Prebiotics and postbiotics have a wide range of applications including immunocompromised patients, autoimmune disease patients, and severe liver or kidney dysfunction patients, while the use of probiotics in above patients will aggravate the patient’s condition [43]. They provide many benefits to the body through a variety of mechanisms, including enhancing digestion, boosting immune function and preventing intestinal infections. However, probiotics are not universally applicable and should be used with caution in certain individuals, such as those with immune system deficiencies or severe liver or kidney dysfunction.

This study focuses on patent searches and aims to explore the global filing trends of prebiotic and postbiotic patents in recent years. Although this study aims to cover as much patents as possible, many factors may lead to missing some patents. Firstly, this study is limited to granted patents and does not include unpublished or under-research patents. This limitation may result in neglect relevant information on ongoing research and patents that still in the research and development stage. Secondly, the research content or technology of the patent is not specific enough, leading to its oversight during patent searches. Thirdly, the patent is from a distant time period, and incomplete preservation in the database results in a less comprehensive patent search. Therefore, this study uses patent search as a tool to predict references of research direction for the study of prebiotics and postbiotics.

## 5. Conclusions and Future Trends

This study is the initial attempt to predict prebiotics and postbiotics utilization directions by exploiting patent-based measures. We analyzed patents from 2001 to 2020 and the most active areas of research were influences on the gut microbiota. Despite not leading in the number of prebiotics and postbiotics patents, China maintains a strong presence in key patent markets. The current expansion of prebiotics and postbiotics presents a golden opportunity for scientific researchers to make significant strides in this field, particularly in the realm of animal nutrition. Based on an analysis of patent mining, the utilization of prebiotics and postbiotics has been predominantly associated with metabolite, fatty acid, polyethylene glycol, and gut microbiota. When consumed by humans and animals, prebiotics and postbiotics induce changes in the gut microbiota structure, resulting in immunological alterations in the gastrointestinal tract. This has significant implications for animal health and nutrition, affecting growth, disease resistance, and overall wellbeing. Given the inherent complexity of the prebiotics and postbiotics domain, these findings point to promising research directions in both human and animal nutrition. However, the mechanism of action for prebiotics and postbiotics remains elusive, underscoring the need for further research. As such, our study could potentially serve as a roadmap for future investigations in this field, particularly those focusing on enhancing animal nutrition and health. In conclusion, the analysis of prebiotics and postbiotics related patents provides valuable insights into the potential future trends in nutritional health. The increasing number of patents in these fields suggests a growing recognition of the importance of gut health in overall wellness, and the potential of these compounds affect the health gut microbiome. Furthermore, the diversity of these patents, ranging from food products to pharmaceutical compositions, indicates the wide-ranging applications of prebiotics and postbiotics in promoting health and preventing disease. As such, patent analysis serves as a useful tool in predicting the future directions of research and development in nutritional health.

## Figures and Tables

**Figure 1 nutrients-16-00380-f001:**
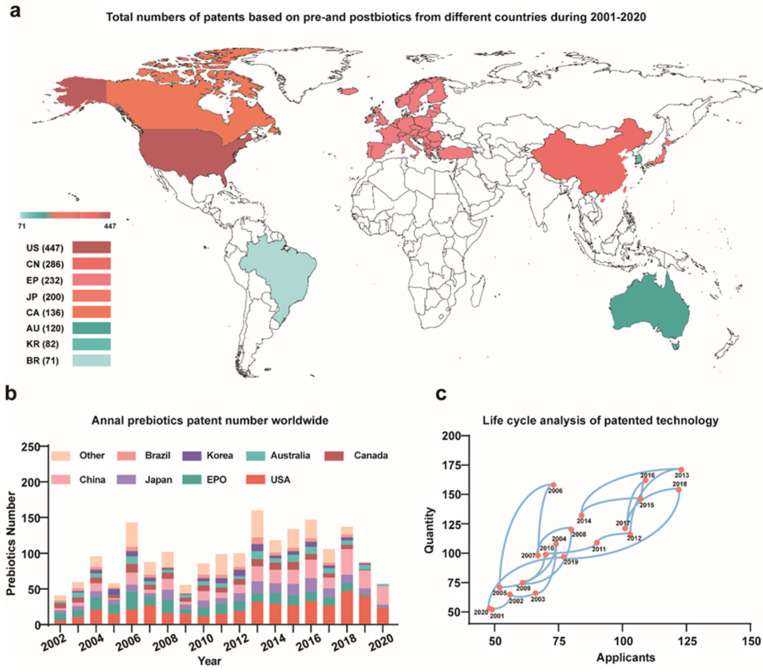
Global distribution of total prebiotic and postbiotic patents from 2001 to 2020. (**a**) Application by country (**b**) Annual patent distribution (**c**) Annual trends of patent applications.

**Figure 2 nutrients-16-00380-f002:**
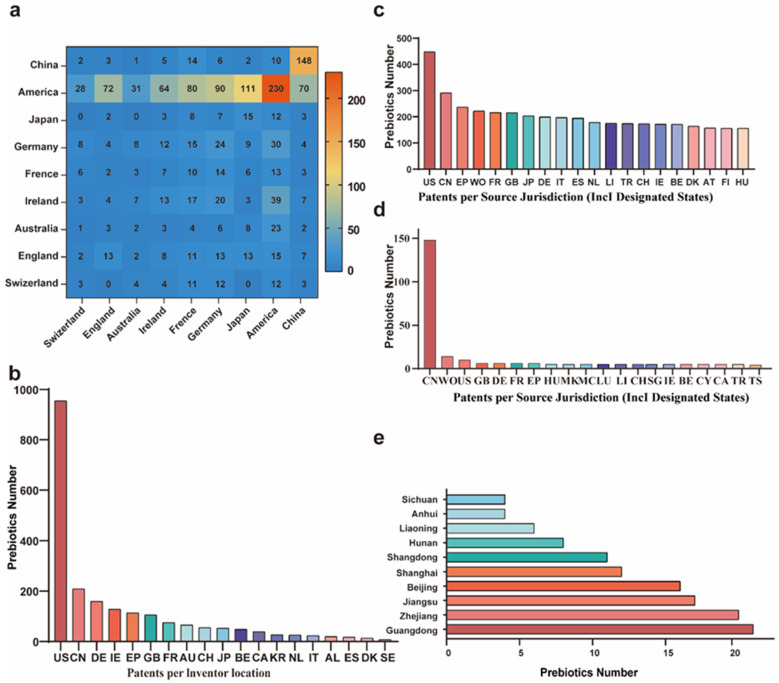
Geographical distribution of prebiotic and postbiotic patent applications. (**a**) Heat map estimated from quantitative data of the patent distribution from the primary countries of patent technologies of prebiotics and postbiotics globally (**b**) Patent application technology; country of origin inventor location (**c**) Technology applications in technologically advanced countries (**d**) Distribution of inventions and creations from China to overseas patents (**e**) Ranking of patent numbers from provinces and cities in China. Abbreviation: American (US), China (CN), Japan (JP), Germany (DE), France (FR), Ireland (IE), Australia (AU), Switzerland (CH), European Patent Office (EP), Spain (ES), Denmark (DK), Great Britain (GB), Belgium (BE), Canada (CA), South Korea (KR), Netherlands (NL), Italy (IT), Albania (AL), Sweden (SE).

**Figure 3 nutrients-16-00380-f003:**
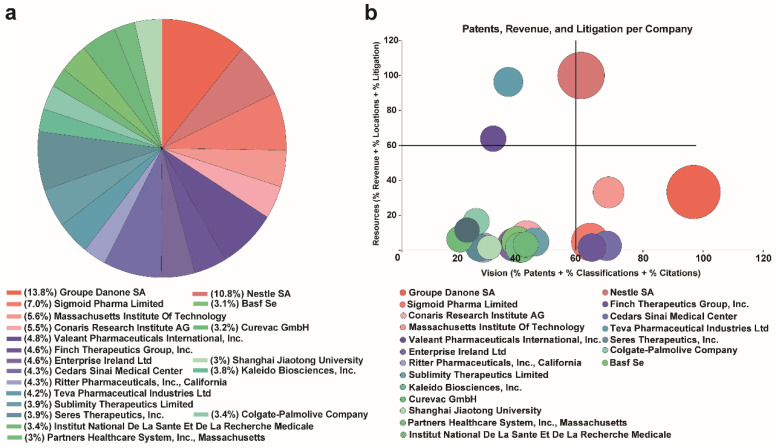
Patent applicant analysis. (**a**) Ranking and distribution of prebiotics and postbiotics applications in global companies and individuals (**b**) Number of patent applications, grants and global marked competitiveness. The *x*-axis represents technology competitiveness, while the *y*-axis represents strong economic strength.

**Figure 4 nutrients-16-00380-f004:**
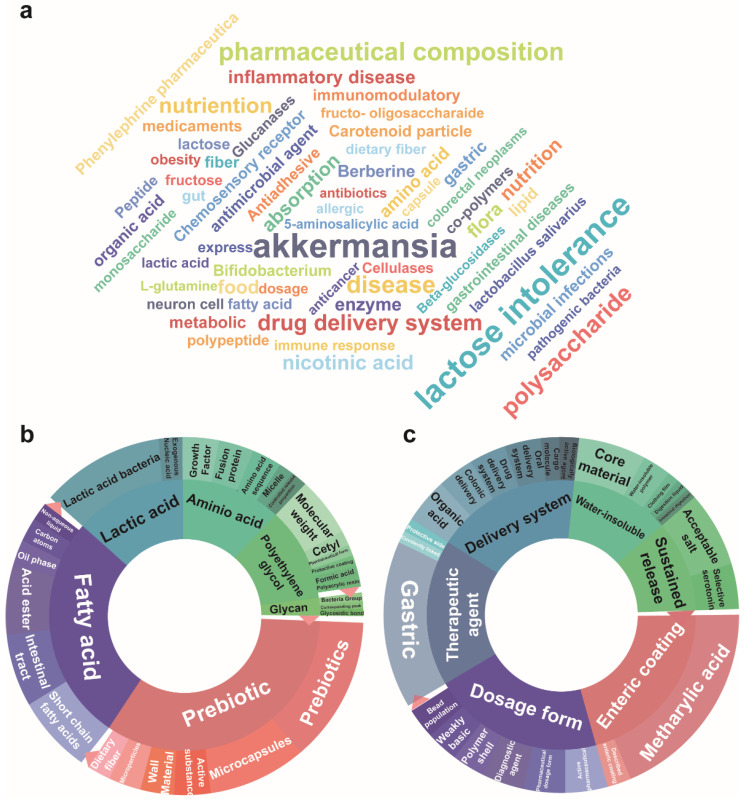
Analysis of prebiotics and postbiotics relevant patents. (**a**) Metabolite- related patents of prebiotics and postbiotics represented by word cloud (**b**) Correlations between metabolites related patents and prebiotics and postbiotics represented by text clustering (**c**) Correlations between delivery material related patents and prebiotics and postbiotics using text clustering.

**Figure 5 nutrients-16-00380-f005:**
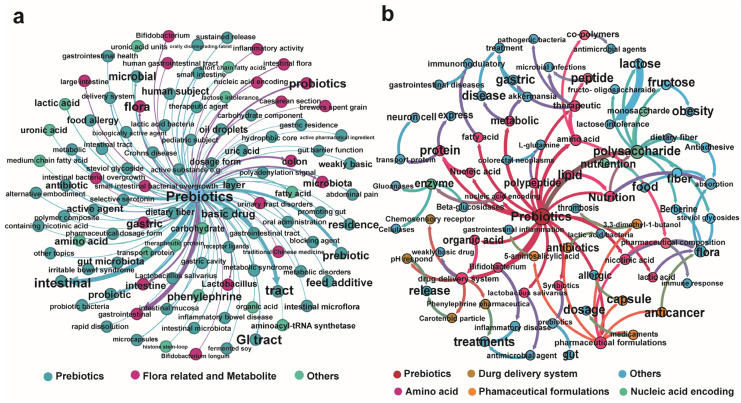
Relevant factor analysis of prebiotics. (**a**) Association between prebiotics and the indicated areas (**b**) Multi-level relationships between prebiotics and postbiotics and other metabolites.

**Table 1 nutrients-16-00380-t001:** IPC numbers related to prebiotic and postbiotic intestinal absorption regulation technologies.

Number	Classification Codes	Description
1	A23K001	Animal feed (Old version)
2	A23K010	Animal feed
3	A23K020	Additional food elements of animal feed
4	A23L031	Edible extracts or preparations of fungi
5	A23L033	Change the nutritional properties of food; nutritional products
6	A61K009	Pharmaceutical preparations characterized by special physical shapes
7	A61K031	Pharmaceutical preparations containing organic active ingredients
8	A61K035	Medical preparations containing raw materials of unknown structure or their reaction products
9	A61K038	Peptide-containing pharmaceutical preparations
10	A61P001	Peptide-containing pharmaceutical preparations
11	A61P003	Drugs for the treatment of metabolic diseases

## Data Availability

No new data were created or analyzed in this study. Data sharing is not applicable to this article.

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
