# Peer review of "The History and Prediction of Prebiotics and Postbiotics: A Patent Analysis"

_nutrients, 2024, doi:10.3390/nu16030380_

Round 1

Reviewer 1 Report

Comments and Suggestions for Authors

The paper titled "The history and prediction of prebiotics and postbiotics: a patent analysis," conducts a comprehensive analysis of patents related to prebiotics and postbiotics. It classifies and examines 2215 patents awarded from 2001 to 2020, providing insights into the development and future directions of these biological agents. Key areas include functional patent classification, global patent application trends, geographical analysis of patent applications, and assessment of patent applicants and market competitiveness.

The research concludes that patent analysis can predict the future trajectories of prebiotics and postbiotics, offering valuable perspectives for assessing their advancement. The study emphasizes the importance of these compounds in human and animal health, and their potential in nutrition and disease prevention. However, it notes the need for more research to fully understand their action mechanisms.

The article provides an exhaustive and detailed analysis of trends in prebiotic and postbiotic research and development, using patent analysis as the primary tool. This study can be a valuable reference for future research in this field, offering a global view and grounded predictions on the future directions of prebiotic and postbiotic use.

Strengths:

- The article demonstrates thoroughness in data collection, utilizing multiple databases to compile a comprehensive set of patents.

- The study's approach of using patent analysis to predict future trends is novel and could be valuable for stakeholders in the field.

- The article offers a global view of the patent landscape, which is crucial for understanding the worldwide impact and development in this area.

However, as a reviewer, I'd like to highlight several considerations:

Major comments:

1. The methodology section could benefit from more detailed explanations of the criteria and processes used for patent selection and classification. Clearer articulation of these methods would enhance the study's reproducibility and reliability.

2. The study focuses on the quantity of patents but does not deeply analyze the quality or the impact of these patents on the field. Including an assessment of patent impact could provide a more nuanced understanding of the field's development.

3. The article should address potential limitations, such as biases in patent databases or the exclusion of relevant patents not captured in the searched databases.

4. While the paper does an excellent job of presenting the data, it could benefit from a more detailed discussion on how these findings fit within the broader scientific literature and what implications they have for future research and application.

5. Inclusion of a Comprehensive Table of Patents: Incorporating a detailed table listing all analyzed patents could significantly enhance the paper's utility and clarity. 

Potential columns for inclusion in the table could be: patent number, title, date of issuance, inventors, assigning entity, geographical jurisdiction, classification of technology, and its relevance to the field. This structure would allow for a comprehensive yet concise overview of each patent, facilitating a deeper understanding of the scope and impact within the field.

The researchers have undertaken an overwhelming task of compilation from which the scientific community cannot benefit due to the lack of this summary. By adding such a table, the paper would not only augment its informational value but also provide a practical tool for researchers, potentially stimulating further study and application in the realm of prebiotics and postbiotics.

However, as a reviewer, I acknowledge the complexity involved in creating such a table, yet I believe the significant work by the authors warrants inclusion in a summary format. Such an addition would elevate this article to a cornerstone paper, widely referenced and cited for its comprehensive data. The extensive efforts by the authors should not be overlooked. If a complete table is impractical, perhaps a curated selection of around 100 key patents could be presented, enabling researchers to leverage the valuable insights garnered by this team's work. Selecting the top 100 patents could be based on criteria such as the impact on the field, citation frequency, novelty, and technological significance.

The inclusion of this list or table is imperative, particularly since the specific patents referred to cannot be identified, as they are not included in the bibliography.

Minor comments:

6. The images, particularly the graphical representations and diagrams, provide a visual summary of the patent analysis. However, their impact could be enhanced with more detailed legends and explanations and it would be advisable to delineate the meanings of all abbreviations utilized in the graphical representations.

7. Some of the figures (e.g., Figure 3 and Figure 5) appear to contain small text and complex data which might be challenging for readers to interpret. Increasing the font size could improve readability.

Comments on the Quality of English Language

8. The phrase "In addition patents for these additives have also increased yet their functional classification have been problematic." should be revised for grammatical accuracy. Suggestion: "In addition, the number of patents for these additives has also increased, yet their functional classification has been problematic."

9. The phrase "a period from 2001 to 2013 where 42 patents were issued annually and a surge exceeding 140 patents annually subsequent to 2013." could be clearer. Suggestion: "a period from 2001 to 2013 during which an average of 42 patents were issued annually, followed by a surge exceeding 140 patents annually after 2013."

10. The phrase "decrease the relative abundance of bacterial pathogens and choline metabolizers" may be more accurately phrased as "decreases the relative abundance of pathogenic bacteria and choline metabolizers."

11. "The survival and stability of probiotics in dairy products are subject to various processing conditions" should be corrected to "The survival and stability of probiotics in dairy products is subject to various processing conditions."

Author Response

Thank you very much for taking the time to review this manuscript. Please find the detailed reponses below and corresponding revisions in track changes in the re-submitted files.

Reviewer 2 Report

Comments and Suggestions for Authors

Dear Authors,

The presented manuscript entitled:The history and prediction of prebiotics and postbiotics: a patent analysis” describes the prediction of  the development direction of prebiotics and postbiotics through patent mining, thereby holding a significant reference value in evaluating development of these compounds. This analysis provides a comprehensive framework for understanding of the development trends and future application fields for prebiotics and postbiotics.

The article has some scientific value and the item under review meets the novelty criterion. A good review must be informative about the field and focus on a topic in a way that has not been done before.  It means that a review article needs to go beyond mere description and ‘state-of-the-literature’ summaries. The presented manuscript shows and develops a new way of thinking about an analyzed topic. Authors also think comprehensively and combine the various issues related to the topic. However,  there are some problems  need to be fixed before the publication.

Firstly, the title of the article includes the word „prediction”. However, it is difficult to address this issue in the text. I think that a good solution is to title the last of subsection no. 4 as : “Conclusions and future trends” and complete there  the discussion on this topic.

The Authors should also tackle the aspect using  of prebiotics and postbiotics in food production. Is the use of postbiotics in food production more beneficial in terms of quality and safety of the final products than the use of selected microorganisms’ strains, starters  with probiotic properties?

The presented manuscript  includes only 31 references. This rightly concerns in the aspect of exhaustion of the discussed topic.

The other, minor suggestions for improvement are as follows:

·         Figure 3: unreadable. Please correct it.

·         Subsection no. 2.1: what timeframe did the data analysis cover?

From my standpoint, this manuscript is appropriate for publication in the Journal – Nutrients,  after major revision, given the above aspects.  

Author Response

Thank you very much for taking the time to review this manuscript. Please find the detailed responses below and the corresponding revisions in track changes in the re-submitted files.

Reviewer 3 Report

Comments and Suggestions for Authors

The review manuscript is skillfully crafted, endeavoring to anticipate the directions of prebiotics and postbiotics utilization through an insightful exploration of patent-based metrics. It offers valuable insights into the trends shaping this field, making it a noteworthy contribution to the current discourse.

Minor parts:

Some descriptions within the figures are not clear, and the font size is insufficient. For instance, the fonts in Figure 3 are generally too small, and there are specific words within Figure 4 that are also inadequately sized.

Author Response

Thank you very much for taking the time to review this manuscript. Please find the detailed reponses below and the in corresponding revisions in track changes in the re-submitted files.

Round 2

Reviewer 1 Report

Comments and Suggestions for Authors

The manuscript has undergone a thorough revision in light of the feedback provided by various reviewers. It is evident that the authors have exerted considerable effort to enhance and incorporate the suggested modifications, particularly in regards to the discussion and limitations sections and to the supplementary table, which will serve as a valuable summary for other researchers. In my specific case, I would only suggest altering the title of the Supplementary Table. Instead of "Table S1. Information on the functional role of prebiotics," it could be more appropriately titled "Table S1. Summary Table of the Key Probiotic Patents Analyzed in this Study. Information on the Functional Role of Prebiotics," or a similar title at the authors' discretion.

Reviewer 2 Report

Comments and Suggestions for Authors

Dear Authors,

I  would like to thank you for your response to my review. All of my  previous suggestions were taken into an account. I have no objections to the revised version of the manuscript.  I accept the new  version of the article in present  form.

From my standpoint, revised version of the  manuscript is appropriate for publication in Journal – Nutrients, given the above aspects.